# TH17 cells promote CNS inflammation by sensing danger signals via Mincle

Quanri Zhang [1], Weiwei Liu[1], Han Wang[1], Hao Zhou[1,11], Katarzyna Bulek[1,2], Xing Chen[1], Cun-Jin Zhang [3], Junjie Zhao[1], Renliang Zhang[4], Caini Liu[1], Zizhen Kang[5], Robert A. Bermel [6], George Dubyak [7], Derek W. Abbott[8], Tsan Sam Xiao [8], Laura E. Nagy [1,9,10,12✉] & Xiaoxia Li [1,12✉]

The C-type lectin receptor Mincle is known for its important role in innate immune cells in recognizing pathogen and damage associated molecular patterns. Here we report a T cell–intrinsic role for Mincle in the pathogenesis of experimental autoimmune encephalomyelitis (EAE). Genomic deletion of Mincle in T cells impairs TH17, but not TH1 cell-mediated EAE, in alignment with significantly higher expression of Mincle in TH17 cells than in TH1 cells. Mechanistically, dying cells release β-glucosylceramide during inflammation, which serves as natural ligand for Mincle. Ligand engagement induces activation of the ASC-NLRP3 inflammasome, which leads to Caspase8-dependent IL-1β production and consequentially TH17 cell proliferation via an autocrine regulatory loop. Chemical inhibition of β-glucosylceramide synthesis greatly reduces inflammatory CD4+ T cells in the central nervous system and inhibits EAE progression in mice. Taken together, this study indicates that sensing of danger signals by Mincle on TH17 cells plays a critical role in promoting CNS inflammation.

[1] Department of Inflammation and Immunity, Cleveland Clinic, Lerner Research Institute, Cleveland, OH, USA. [2] Department of Immunology, Faculty of Biochemistry, Biophysics, and Biotechnology, Jagiellonian University, Krakow, Poland. [3] Department of Neurology, Nanjing Drum Tower Hospital, Medical School and the State Key Laboratory of Pharmaceutical Biotechnology, Nanjing University, Nanjing, Jiangsu, China. [4] Proteomics and Metabolomics Core, Department of Research Core Services, Lerner Research Institute, Cleveland, OH, USA. [5] Department of Pathology, University of Iowa, Iowa, IA, USA. [6] Mellen Center for Multiple Sclerosis, Cleveland Clinic, Cleveland, OH, USA. [7] Department of Physiology and Biophysics, University Hospitals Cleveland Medical Center, Case Western Reserve University School of Medicine, Cleveland, OH, USA. [8] Department of Pathology, Case Western Reserve University, Cleveland, OH, USA. [9] Department of Gastroenterology and Hepatology, Cleveland Clinic, Cleveland, OH, United States. [10] Department of Molecular Medicine, Case Western Reserve University, Cleveland, OH, United States. [11]Present address: Division of Transplant Surgery, Department of Surgery, Brigham and Women's Hospital, Harvard Medical School, Boston, MA 02115, USA. [12]These authors jointly supervised this work: Laura E. Nagy, Xiaoxia Li. ✉email: nagyl3@ccf.org; lix@ccf.org

Multiple sclerosis (MS) is an inflammatory demyelinating disease of the central nervous system (CNS)[1–3]. Numerous studies indicate that the inflammatory process in MS and experimental autoimmune encephalomyelitis (EAE) is initiated by autoreactive CD4+ T cells that are reactive against myelin[4,5]. During the initiation stage of EAE, antigen-presenting cells (APCs) produce cytokines that regulate the differentiation of effector CD4+ T cells, polarizing these cells to $T_H1$ (producing IFNγ) and $T_H17$ (producing IL-17) T-cell lineages[6,7]. Previous studies have reported the critical roles of pattern recognition receptors on APCs in autoimmune inflammatory responses[8,9]. While Toll-like receptors (TLRs) are well known for their ability in modulating $T_H1$ and $T_H17$ responses[10,11]; C-type lectin receptors (CLRs) have also begun to take a center stage in T-cell-mediated autoimmune diseases, including MS and EAE[12–14]. Notably, type II transmembrane CLRs carry a carbohydrate-recognition domain; this family of CLRs included Dectin-1, Dectin-2, and macrophage-inducible C-type lectin (Mincle). These CLRs are important immune modulators through the recognition of pathogen-associated molecular patterns and damage-associated molecular patterns (DAMPs). The activation of CLR signaling activates APCs, enabling the differentiation of CD4+ IL-17-producing effector T cells ($T_H17$ cells) during host defense against fungal infection and pathogenesis of autoimmune diseases such as MS and EAE[15–18]. In addition to the indirect roles of TLRs and CLRs in promoting T-cell differentiation through DC maturation and production of regulatory cytokines, emerging evidence indicates that TLR signaling via TLR2 and TLR4 is activated in CD4+ T cells to promote cytokine secretion or modulate their function[19,20]. Considering the robust impact of CLRs on $T_H17$ responses, it is critical to examine the possible expression of CLRs on CD4+ T cells and determine whether they have any direct role in promoting $T_H17$ cells.

Here we report a T cell-intrinsic Mincle-mediated inflammasome activation that results in IL-1β production critical for $T_H17$-mediated EAE pathogenesis. Unexpectedly, we observed that Mincle was highly expressed in polarized $T_H17$ cells, but not $T_H1$ cells. $T_H17$ polarizing cytokines IL-1 and IL-6 induced the expression of Mincle in CD4+ cells. We observed that T-cell-intrinsic Mincle was required for the effector stage of EAE, and Mincle deficiency in T cells impaired $T_H17$, but not $T_H1$, cell-mediated EAE. Mechanistically, Mincle activation (by endogenous and exogenous ligands) drove the activated $T_H17$ cells to produce IL-1β via ASC-NLRP3–dependent caspase-8 activation. While IL-1 receptor (IL-1R) was specifically expressed on $T_H17$ cells but not on $T_H1$ cells, Mincle-activated $T_H17$ cells exhibited enhanced cell proliferation in an IL-1R-dependent manner, suggesting autocrine action of Mincle-promoted $T_H17$-derived IL-1β. Interestingly, Mincle endogenous ligand, β-glucosylceramide released by dying cells, promoted $T_H17$ cell proliferation in a Mincle-dependent manner; blockade of β-glucosylceramide synthesis rescued the mice from EAE. Lipids extracted from the spinal cord of EAE mice promoted $T_H17$ cell expansion, whereas lipid extracts from the spinal cord of mice treated with glucosylceramide synthase inhibitor-AMP-DNM failed to promote $T_H17$ cells expansion.

## Results

**Mincle is highly expressed in $T_H17$ cells.** Mincle is well known as an inducible receptor on innate immune cells, including macrophages[21]. However, little is known regarding the expression and function of Mincle in T cells. Interestingly, we found that Mincle mRNA and protein were highly elevated in $T_H17$ cells, but not in $T_H0$, $T_H1$, $T_H2$ or Treg cells (Fig. 1a, b). In the presence of anti-CD3 and anti-CD28, Mincle mRNA expression was highly induced in CD4 T cells by IL-6, and to a lesser extent by IL-1β and IL-23 (Fig. 1c). Notably, Mincle deficiency had no impact on the polarization of $T_H1$, $T_H2$, $T_H17$, or Treg cells (Fig. 1d). Furthermore, we reanalyzed three RNA seq datasets of CD4 or $T_H17$ cells from EAE mice[22–24] and found that Mincle is among the highly expressed CLRs genes in splenic CD4 T cells and as well as in CNS $T_H17$ cells (Supplemental Fig. 1a–c). Similar to the increased expression of Mincle in $T_H17$, other members of CLR family, including Dectin-1 (*Clec7a*), Dectin-2 (*Clec4n*), and Dectin-3 (*Clec4d*, MCL), were also induced in $T_H17$. However, this response was not affected by Mincle deficiency (Supplemental Fig. 1d). Dectin-3/MCL is required for induction of Mincle in response to stimulation by TDM (trehalose-6,6-dimycolate)[13]. Further, our data showed that Mincle is only abundantly expressed at the late stages of in vitro polarization and in vivo priming (Supplemental Fig. 1e). Therefore, we tested MCL was also essential for the induction of Mincle in $T_H17$ cells. However, expression of *Clec4d* mRNA in $T_H17$ was much lower compared to other CLRs and Dectin-3/MCL protein was not detectable by western blot (Supplemental Fig. 1d, f). Taken together, these data indicate that Mincle is likely induced in $T_H17$ cells in an MCL-independent manner.

**T-cell-specific deficiency of Mincle delays and reduces EAE.** To investigate whether Mincle has a T cell-intrinsic role, we crossed a mouse strain in which exon 3–5 of the gene *Clec4e* (which encodes Mincle) (*Mincle$^{f/f}$* mice) is flanked by loxP sites onto the *Lck-Cre* transgenic mouse strain, which expresses Cre under the control of the *Lck* proximal promoter, generating *Mincle$^{f/f}$Lck-Cre* and *Mincle$^{f/+}$Lck-Cre* mice (Fig. 2a and Supplementary Fig. 2a, b). Mincle expression was efficiently and specifically deleted on T cells isolated from *Mincle$^{f/f}$Lck-cre* mice (Fig. 2a). We then tested the impact of T-cell-specific *Mincle* deletion on neuroinflammation and demyelination by immunizing *Mincle$^{f/f}$Lck-Cre* and littermate control *Mincle$^{f/+}$Lck-Cre* with the neuroantigen MOG35-55 peptide. Mice with T-cell-specific *Mincle* deficiency had attenuated disease severity compared with controls (Fig. 2b). As a control, we showed that *Mincle$^{+/+}$* and *Mincle$^{f/f}$* mice developed comparable EAE disease, indicating that the floxed allele did not affect the development of EAE (Supplementary Fig. 2c). Inflammatory mononuclear cell infiltration in the brain, including CD4+ T cells, neutrophils, and macrophages, was substantially reduced in mice with T-cell-specific *Mincle* deletion compared with controls (Fig. 2c, d), and the expression of inflammatory cytokines and chemokines in the spinal cord was also significantly decreased (Fig. 2e). Histopathological analysis showed substantially reduced accumulation of infiltrating immune cells and demyelination in spinal cords of *Mincle$^{f/f}$Lck-Cre* mice than that in littermate control *Mincle$^{f/+}$Lck-Cre* mice (Fig. 2f). Further analysis of infiltrating CD4+ T cells showed that T-cell-specific *Mincle* deficiency resulted in a reduction of pathogenic $T_H17$ cells (IL-17A+, GM-CSF+), but not IL-10 and IFN-γ producing CD4+ T cells (Fig. 2g). Likewise, *Mincle$^{f/f}$CD4-cre* exhibited substantially reduced EAE disease severity compared to the littermate control *Mincle$^{f/+}$CD4-Cre* mice (Supplementary Fig. 2d). Infiltrating immune cells and demyelination were also dramatically reduced in the spinal cord of *Mincle$^{f/f}$CD4-cre* mice (Supplementary Fig. 2e, f). Unexpectedly, deletion of *Mincle* in myeloid cells or microglia had little impact on EAE pathogenesis (Supplementary Fig. 2g, h). Taken together, these data suggest that T-cell-intrinsic *Mincle* plays a critical role in the pathogenesis of EAE.

**T-cell-specific *Mincle* deficiency did not affect T-cell priming.** Since the EAE phenotype was greatly reduced in *Mincle$^{f/f}$Lck-Cre* and *Mincle$^{f/f}$CD4-Cre* mice after immunization with MOG35-55

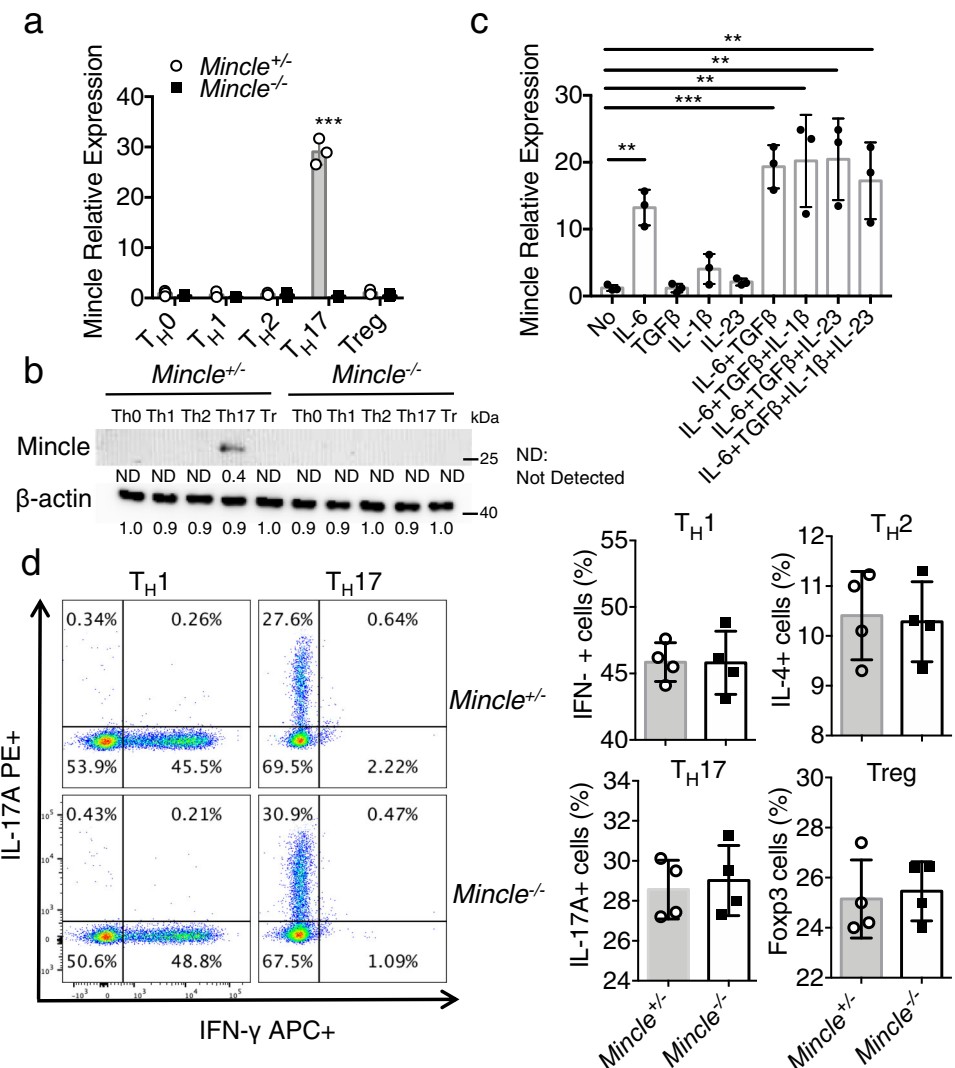

**Fig. 1 Mincle is specifically expressed in T$_H$17 cells. a** Real-time PCR analysis for mRNA levels of *Mincle* in T$_H$0, T$_H$1, T$_H$2, T$_H$17, and Treg cells after 3 days polarization. The expression levels were normalized to the expression of β-actin, $n = 3$ biological replicates. **b** Western analysis of Mincle protein in polarized T$_H$1, T$_H$2, T$_H$17, and Treg cells from WT and *Mincle*-deficient mice, β-actin as a loading control, data are representative of three independent experiments, density values measured using Image J for the representative blot shown, ND not detected. **c** Real-time PCR analysis for *Mincle* mRNA levels in CD4+ cells stimulated with anti-CD3/CD28 in the presence of indicated cytokines for 72 h, $n = 3$ biological replicates. **d** Flow cytometry analysis of wild-type and *Mincle*-deficient T$_H$1, T$_H$1, T$_H$17, and T$_{reg}$ cells with the indicated antibodies, $n = 4$ biological replicates. ***$P < 0.001$ (Two sided student's *t* test for **a** and **d**, Two-way ANOVA for **c**) Data are represented as mean ± SD. Exact *P* values for asterisks (from left to right): **a** 0.00004, **c** 0.0015, 0.0007, 0.0089, 0.0056, 0.0086.

peptide, we next examined the importance of Mincle for the priming of MOG35-55-reactive effector T-cell populations in secondary lymphoid organs. Notably, IL-17A, IFN-γ and GM-CSF cytokine production from the culture of MOG restimulated lymph node cells were similar from *Mincle$^{f/f}$Lck-Cre* mice to those in controls (Fig. 3a–c and Supplementary Fig. 3a). Further characterization of draining lymph nodes on day 9 post MOG immunization revealed similar CD4+ T-cell activation, proliferation and cytokine production, suggesting that peripheral T-cell priming was not affected in *Mincle$^{f/f}$Lck-cre* mice (Supplementary Fig. 3b–e). These results are consistent with the ex vivo polarization experiments, which showed that *Mincle* deficiency had no impact on the polarization of T$_H$1, T$_H$2, T$_H$17 or Treg cells. Taken together, these results indicate that T-cell-specific *Mincle* deficiency had no impact on ex vivo T-cell differentiation or primary MOG35-55-specific T-cell priming in vivo.

**T-cell-intrinsic *Mincle* is required for T$_H$17, but not T$_H$1, -mediated EAE.** The fact that T-cell-specific *Mincle* deficiency attenuated EAE pathogenesis but not T-cell priming, promoted us to examine the pathogenic role of *Mincle*-deficient T$_H$1 and T$_H$17 cells via adoptive transfer into naive recipients. Wild-type recipients receiving T$_H$1 cells from *Mincle$^{f/f}$Lck-Cre* mice developed a similar diseases as those receiving T$_H$1 cells from *Mincle$^{f/+}$Lck-Cre* mice (Fig. 3d). But wild-type mice receiving T$_H$17 cells from *Mincle$^{f/f}$Lck-Cre* mice developed disease with reduced severity compared with that in mice receiving T$_H$17 cells from *Mincle$^{f/+}$Lck-Cre* mice (Fig. 3e). Flow cytometry analysis of the infiltrating mononuclear cells in the brains showed that the numbers of CD4+ T cells, macrophages, and neutrophils were also reduced in mice receiving *Mincle*-deficient T$_H$17 cells compared with controls (Fig. 3f). Likewise, histopathological analysis revealed reduced inflammatory-cell infiltration, accompanied by reduced demyelination in mice receiving *Mincle*-deficient T$_H$17 cells (Fig. 3g). Together, these findings indicate

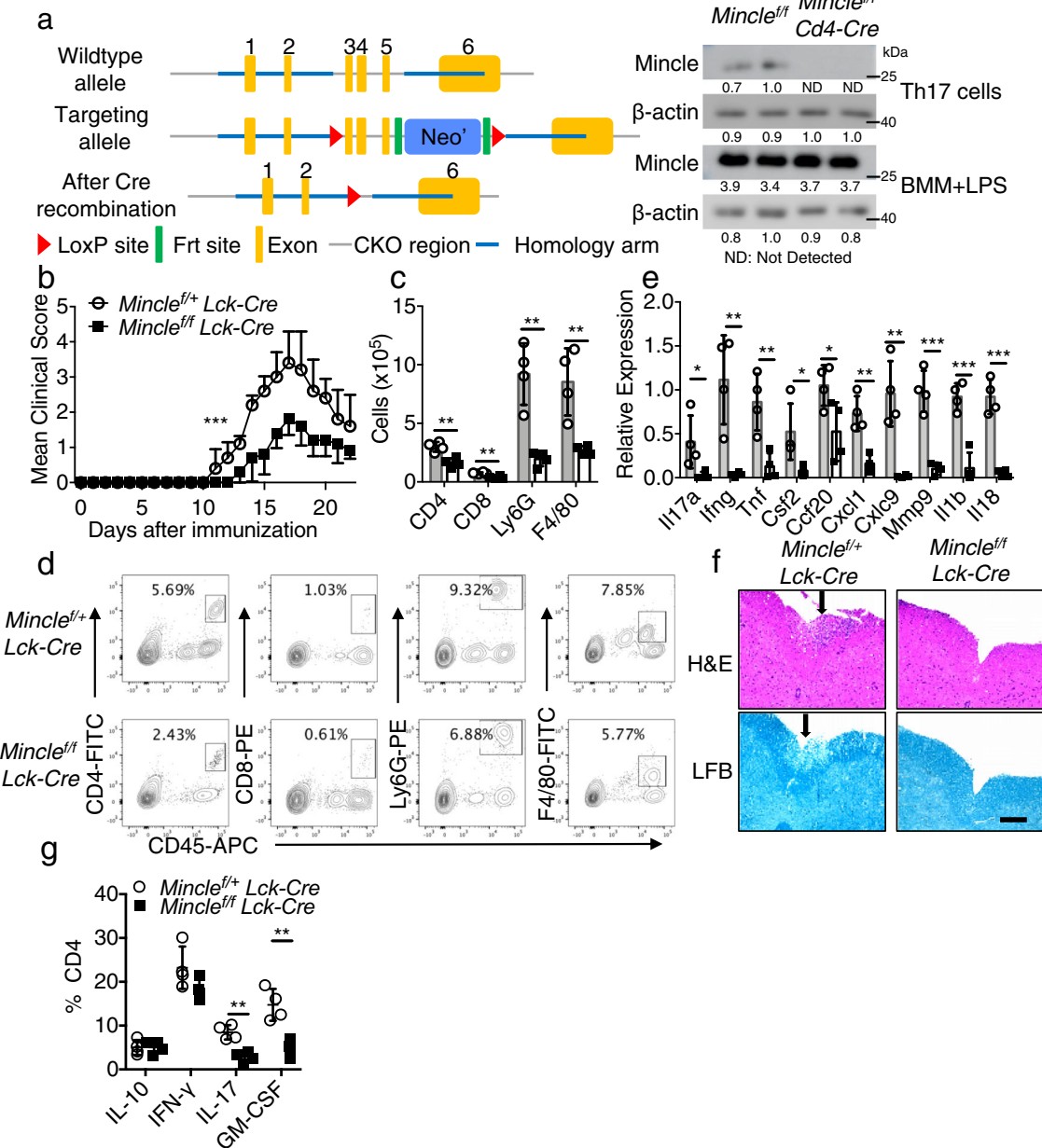

**Fig. 2 T-cell-specific *Mincle* deletion protects mice from EAE. a** Targeting vector design for the generation of a mouse strain with flanking *Clec4e* exon 3-5 by loxP sites and western analysis of *Mincle* protein expression in TH17 cells and bone marrow macrophages (1 µg/ml LPS, 6 h) from *Mincle*[f/+]*Lck-Cre* and *Mincle*[f/f]*Lck-Cre* mice, n = 2 for each genotype, density values measured using Image J for the representative blot shown, ND not detected. **b** Mean clinical score of EAE in *Mincle*[f/+]*Lck-Cre* and *Mincle*[f/f]*Lck-Cre* mice (n = 6 mice in each group) induced by active immunization with MOG$_{35-55}$. **c, d** Absolute cell numbers (**c**) and gating strategy (**d**) of CNS-infiltrating cells were measured at the peak of disease by analyzing brain mononuclear infiltrating cells through flow cytometry with indicated antibodies, n = 4 biological replicates. **e** Real-time PCR analysis of relative mRNA expression of inflammatory genes in the spinal cord from *Mincle*[f/+]*Lck-Cre* and *Mincle*[f/f]*Lck-Cre* mice at the peak of disease. Expression was normalized to β-actin mRNA, n = 4 biological replicates. **f** Hematoxylin and eosin (H&E) staining (upper panels) and Luxol fast blue staining (lower panels) of lumbar spinal cords from *Mincle*[f/+]*Lck-Cre* and *Mincle*[f/f]*Lck-Cre* mice harvested at the peak of disease, Scale bars represent 100 µm. Arrows in the upper panel indicate inflammatory cells infiltration, and arrows in the lower panel indicate demyelination area. Representative data are shown for n = 4. **g** Flow cytometry analysis of infiltrated cytokine-producing CD4 T cells in CNS at the peak of disease, n = 4 biological replicates. *P < 0.05, **P < 0.01 (Two-sided student's t test, **c, e**). *P < 0.05 (Two-way ANOVA for **b**). Data are represented as mean ± SD. Exact P values for asterisks (from left to right): **b** 0.0002 **c** 0.0032 0.0012 0.0017 0.0068 **d** 0.0446 0.0053 0.0062 0.0341 0.0410 0.0021 0.0023 0.0003 0.0005 0.0001 **g** 0.0014 0.0030.

that T-cell-intrinsic *Mincle* is required for T$_H$17, but not T$_H$1, cell-mediated EAE.

To test the effect of *Mincle* signal on T$_H$17 migration and survival, we transferred MOG-reactivated T$_H$17 cells from *Mincle*[f/f]*Lck-Cre* and *Mincle*[f/+]*Lck-Cre* into *Rag1*[−/−] mice. Following T$_H$17 cell transfer, we examined the CD4+ cells in the

peripheral (spleen) and the CNS (brain) on days 3, 6, and day 10. T$_H$17 cells from mice with T-cell-specific *Mincle* deficiency egressed from the spleen and migrated to the CNS normally on days 3 and 6 post-transfer (Fig. 3h, i). However, by day 10 after adoptive transfer, mice with T$_H$17 cells from *Mincle*[f/f]*Lck-Cre* mice had fewer CD4+ T cells in the CNS than did mice with T$_H$17 cells

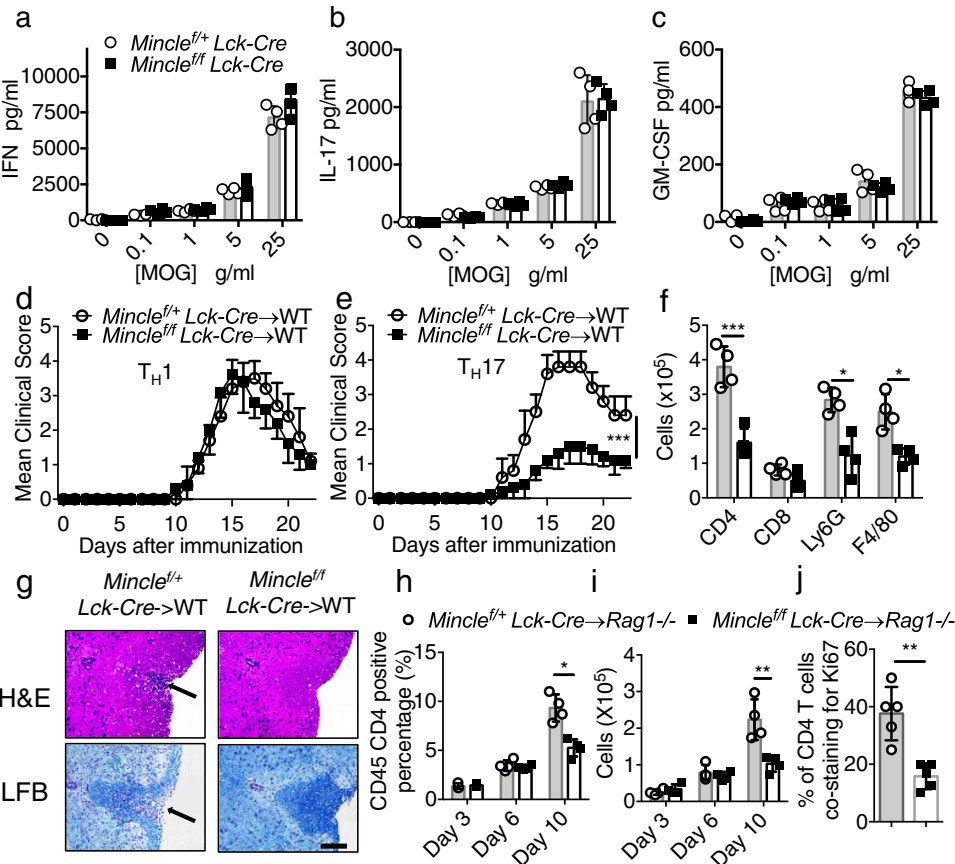

**Fig. 3 *Mincle* is required for $T_H17$ mediated EAE progression. a–c** Lymph nodes were harvested from MOG immunized mice on Day 9 post immunization, and cells were cultured with increasing concentrations of MOG for 72 h. Cytokine concentrations in the culture media were measured by ELISA, $n = 4$ biological replicates. **d, e** Mean clinical score of EAE mice ($n = 5$) induced by adoptive transfer of MOG-reactive **d** $T_H1$ or **e** $T_H17$ cells. **f** Brain lymphocytes from $T_H17$ recipient mice were harvested at the peak of disease and analyzed by flow cytometry with indicated antibodies by flow cytometry, $n = 5$ biological replicates. **g** H&E staining and Luxol fast blue staining of lumbar spinal cords at peak of the disease in recipient mice. Scale bar, 100 μM. Arrows in the upper panel indicate inflammatory cells infiltration, and arrows in the lower panel indicate demyelination area. Representative data are shown for $n = 4$. **h** Percentage of CD45+ CD4+ cells of total CNS infiltrated cells from $Rag1^{-/-}$ mice that received MOG-reactive $T_H17$ cells, $n = 4$ biological replicates. **i** Cell numbers of CNS-infiltrating CD4+ T cells from (**h, j**). Percent of Ki67/CD4 double-positive cells from spinal cords from adoptive transfer experiments 9 days after transfer (representative images shown in Supplemental Fig. 3f.). $n = 5$, biological replicates. **P < 0.01 (Two-sided student's t test for **a–c, f, h–j**). ***P < 0.001 (two-way ANOVA for **d, e**). Data are represented as mean ± SD. Exact P values for asterisks (from left to right): **e** <0.0001 **f** 0.0009 0.0033 0.0037 **h** 0.0025 **i** 0.0069 **j** 0.0015.

from *Mincle^f/+^Lck-Cre* mice. Taken together, these results suggest that *Mincle* is required for $T_H17$ cell survival and/or expansion in the CNS. Consistent with this hypothesis, staining of the proliferation marker Ki67 in the spinal cord revealed more proliferating/infiltrating CD4+ T cells after adoptive transfer of wild-type $T_H17$ cells compared to mice receiving *Mincle*-deficient $T_H17$ cells (Fig. 3j and Supplementary 3f).

**β-glucosylceramide, a Mincle ligand, promotes $T_H17$ cell proliferation.** Since our data indicated a critical role of *Mincle* for in vivo $T_H17$ cell function, we next investigated whether and how Mincle activation may impact $T_H17$ cells ex vivo. Mincle senses divergent ligands released by non-self-pathogenic microbiome and self-ligands released from dying cells[21]. Recent studies have shown that the intracellular metabolite β-glucosylceramide is an endogenous Mincle ligand possessing immunostimulatory activity[25]. To test the potential impact of β-glucosylceramide on $T_H17$ cells, we cultured the $T_H17$ cells with soluble anti-CD3 and anti-CD28 on plates coated with β-glucosylceramide in the presence of IL-6 and TGFβ. Polarizing $T_H17$ cells in the presence of β-glucosylceramide resulted in a higher percentage of IL-17A+ cells (Fig. 4a). Next, the

polarized $T_H17$ cells were labeled with carboxyfluorescein diacetate succinimidyl ester (CFSE) to examine the potential impact of β-glucosylceramide on $T_H17$ cell division. We indeed found that Mincle activation with β-glucosylceramide promoted $T_H17$ cell proliferation (Fig. 4b). Importantly, the impact of β-glucosylceramide on $T_H17$ cells was impaired in *Mincle*-deficient $T_H17$ cells (Fig. 4a, b). Similarly, challenge with another Mincle ligand, trehalose-6,6-dihehenate (TDB), a synthetic analog of mycobacterial tuberculosis cord factor- trehalose-6,6-dimycolate (TDM), to the $T_H17$ cells under polarizing conditions also resulted in a higher percentage of $T_H17$ cells in a *Mincle*-dependent manner (Supplementary Fig. 4a). In addition to enhanced cell proliferation, Mincle activation promoted inflammatory gene expression in $T_H17$ cells, including increased expression of *Csf2, Tnf,* and *Ifng* mRNA (Fig. 4c). β-glucosylceramide stimulation also induced robust induction of Gm-csf+ $T_H17$ cells; this response was substantially reduced by *Mincle* deficiency (Supplementary Fig. 4b). Heat killed, *Mycobacterium Tuberculosis* (HK-Mtb), an adjuvant commonly used for the induction of EAE, contains abundant immunostimulatory ligands, including the Mincle ligand trehalose-6,6'dimycolate (TDM). To evaluate whether there could be a direct effect of HK-Mtb on $T_H17$ cells, we polarized $T_H17$ cells in the presence or

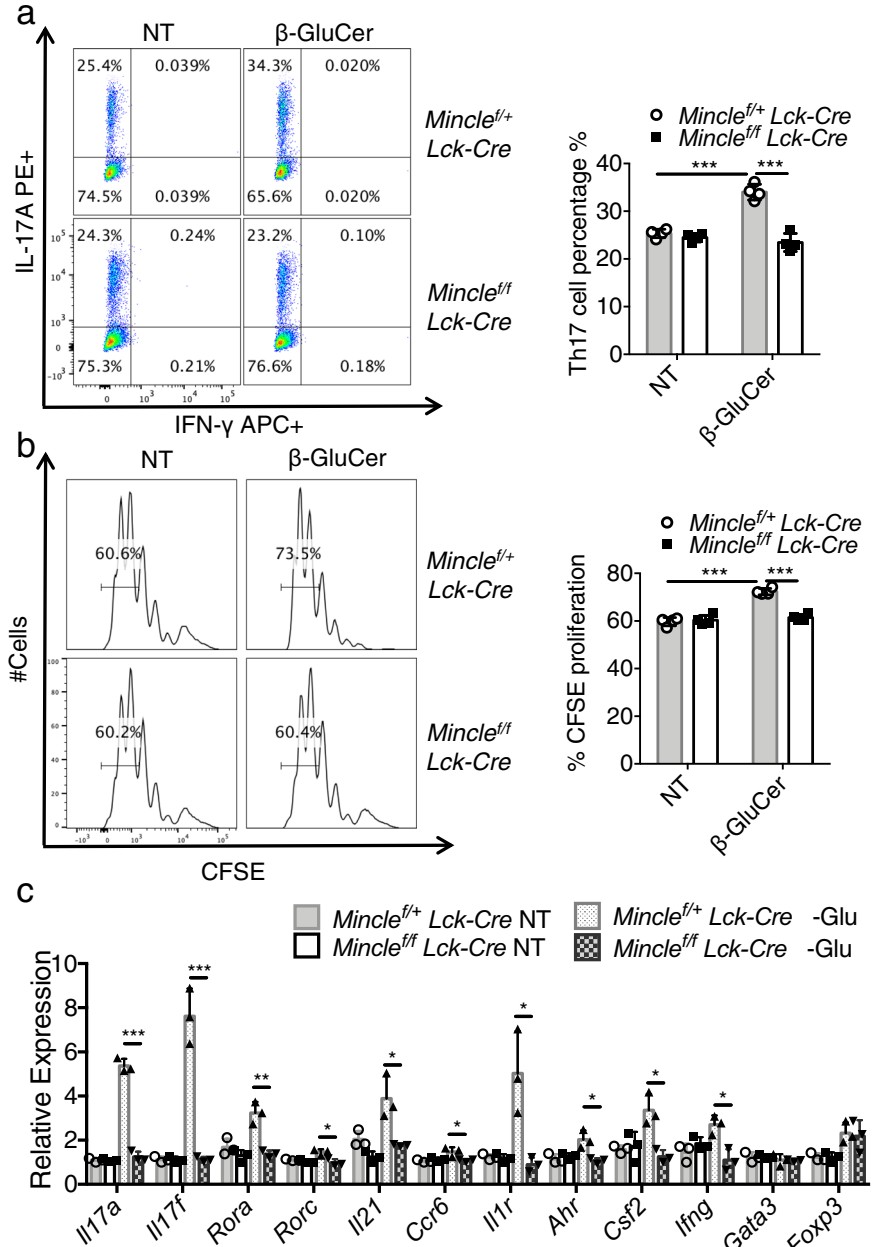

**Fig. 4 Mincle activation promotes T$_H$17 differentiation. a** Flow cytometric analysis of IL-17A and IFNγ from *Mincle$^{f/+}$Lck-Cre* and *Mincle$^{f/f}$Lck-Cre* T$_H$17 polarized with or without β-glucosylceramide (5 µg/ml) stimulation, $n = 4$ biological replicates. **b** CFSE staining of T$_H$17 cells polarized with or without β-glucosylceramide (5 µg/ml). Data are presented as mean fluorescent intensity, $n = 4$ biological replicates. **c** Real-time PCR of mRNA of inflammatory genes in T$_H$17 cells polarized with or without β-glucosylceramide (1 µg/ml), $n = 3$ biological replicates. Expression was normalized to expression of β-actin. *$p < 0.05$, **$p < 0.01$ ***$p < 0.001$ (Two sided student's t test). Data are represented as mean ± SD. Exact $P$ values for asterisks (from left to right): **a** <0.0001 0.00016 **b** <0.0001< 0.0001 **c** 0.00005 0.00088 0.0036 0.0172 0.0225 0.0341 0.0207 0.0178 0.0101 0.0128.

absence of HK-Mtb. Surprisingly, HK-Mtb increased T$_H$17 polarization in both *Mincle$^{f/+}$Lck-Cre* and *Mincle$^{f/f}$Lck-Cre* cells (Supplementary Fig. 4c), indicating that additional Mincle-independent mechanisms are likely involved in the recognition of HK-Mtb by T$_H$17 cells. Taken together, these findings suggest that β-glucosylceramide might activate T$_H$17 cells via Mincle signaling to promote inflammatory T$_H$17 cells, but HK-Mtb has little impact on the activation of Mincle signaling in T$_H$17 cells.

**Mincle activation promotes T$_H$17 cell proliferation via the production of mature IL-1β.** Mincle activation in myeloid cells

results in the secretion of mature IL-1β in LPS-primed condition[26,27]. Furthermore, we previously reported that IL-1β produced by T$_H$17 acts in an autocrine manner on T$_H$17 cells to promote inflammation in CNS[28]. Thus, we next asked whether activation of Mincle on T$_H$17 cells would also promote IL-1β processing and secretion. β-glucosylceramide stimulation indeed induced IL-1β production in T$_H$17 cells; this response was reduced in *Mincle*-deficient T$_H$17 cells (Fig. 5a). β-glucosylceramide-induced IL-1β production was dependent on ASC-NLRP3 inflammasome (Fig. 5b–d). Interestingly, we noted that β-glucosylceramide induced the cleavage and activation of caspase 8, in T$_H$17 cells (Fig. 5b–f). Furthermore, β-glucosylceramide-induced IL-1β

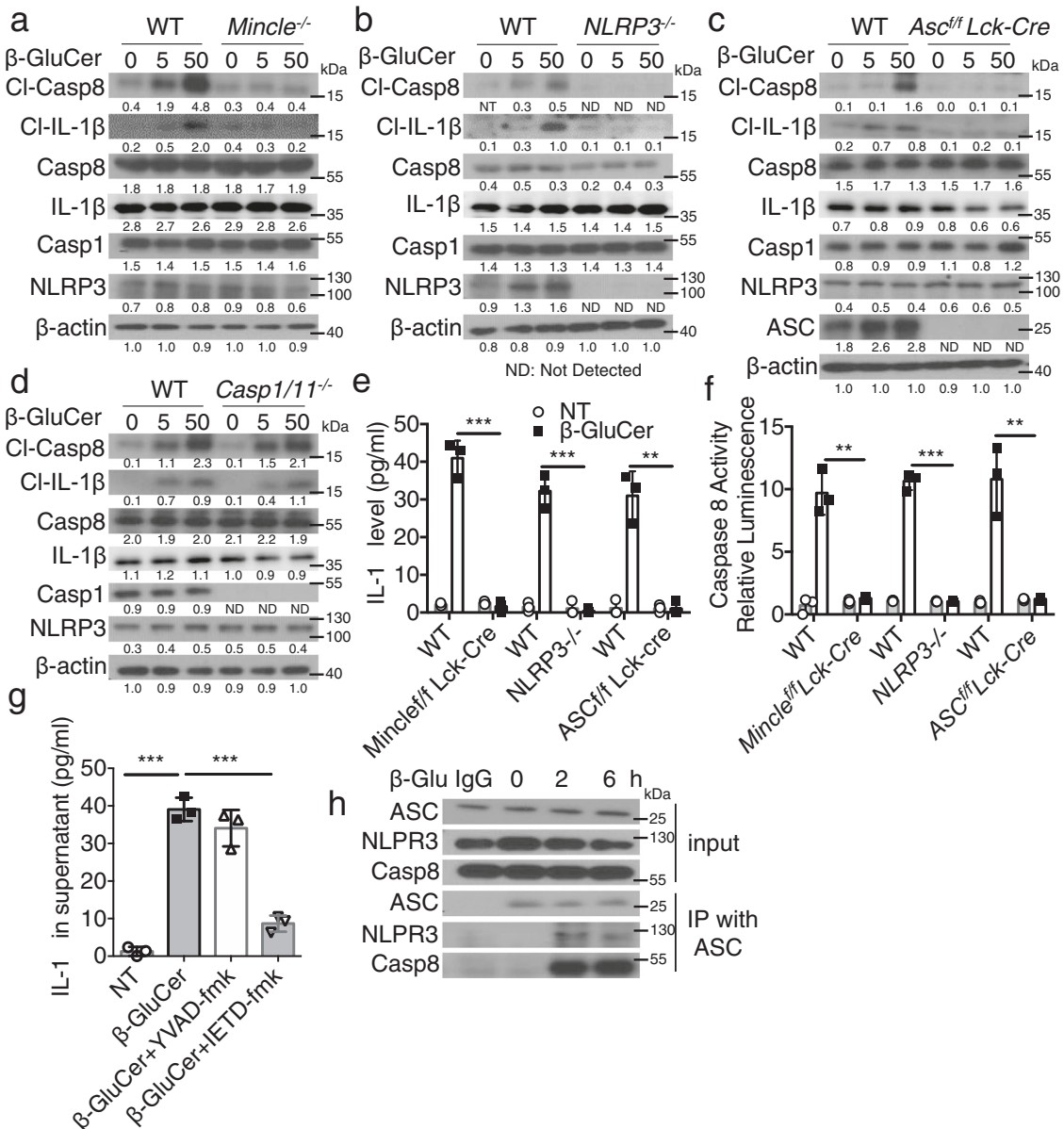

**Fig. 5 Mincle activation leads to pro-IL-1β processing and secretion in T$_H$17 cells. a–d** Polarized T$_H$17 cells from indicated murine strains were stimulated with β-glucosylceramide (0, 5, 50 μg/ml) for 12 h, followed by western blot analysis of supernatants and cell lysates with the indicated antibodies, density values measured using Image J for the representative blot shown, ND not detected. **e** Supernatants from **a–d** were harvested and IL-1β concentrations were determined by ELISA, $n = 3$ biological replicates. **f** T$_H$17 cells were stimulated with β-glucosylceramide (50 μg/ml) for 12 h, caspase 8 Glo Assay reagent was added to the media for another 1 h, followed by analysis by luminescence, $n = 3$ biological replicates. **g** IL-1β concentrations were analyzed from the supernatant of polarized T$_H$17 cells pretreated with caspase inhibitors (YVAD-fmk/caspase1 inhibitor, IETD-fmk/caspase 8 inhibitor) and stimulated with β-glucosylceramide, $n = 3$ biological replicates. **h** Cell lysates from T$_H$17 treated with β-glucosylceramide (50 μg/ml) were subjected to immunoprecipitation with anti-ASC, followed by western analysis with the indicated antibodies, data is representative of three independent experiments. *$p < 0.05$, ***$p < 0.001$ (two-sided student's $t$ test for **e–g**). Data are represented as mean ± SD. Exact $P$ values for asterisks (from left to right): **e** 0.00015 0.00024 0.0015 **f** 0.0011 0.00002 0.0039 **g** <0.0001 0.0002.

production was blocked by caspase 8 inhibition (Fig. 5g) and formation of an ASC-NLRP3-caspase 8 complex was detected upon the β-glucosylceramide stimulation (Fig. 5h).

We next investigated the critical question as to the potential link between Mincle-dependent IL-1β production and T$_H$17 cell proliferation. Using T$_H$17 cells from wild-type and IL-1β-deficient mice, we noted that IL-1β deficiency abolished β-glucosylceramide-induced expansion of inflammatory T$_H$17 cells (Supplementary Fig. 5a). Stimulation of T$_H$17 cells with β-glucosylceramide increased the association of two key downstream mediators of Mincle signaling, FcRγ and Syk kinase, with Mincle (Supplementary Fig. 5b). Although canonical caspase 8 activation leads to cell death,

activation of Mincle signaling did not affect the viability of T$_H$17 cells (Supplementary Fig. 5c). Taken together, these data demonstrate that β-glucosylceramide activated a Mincle-FcRγ-Syk and ASC-NLRP3-caspase 8-dependent IL-1β production and enhanced the proliferation of T$_H$17 cells.

**β-glucosylceramide accumulation during EAE development aggravates the disease.** Sensing of DAMPs released by damaged or dying cells elicits and promotes sterile inflammatory response in the CNS in models of MS and EAE[29]. β-glucosylceramide is released by dying cells[25], which may activate Mincle expressing

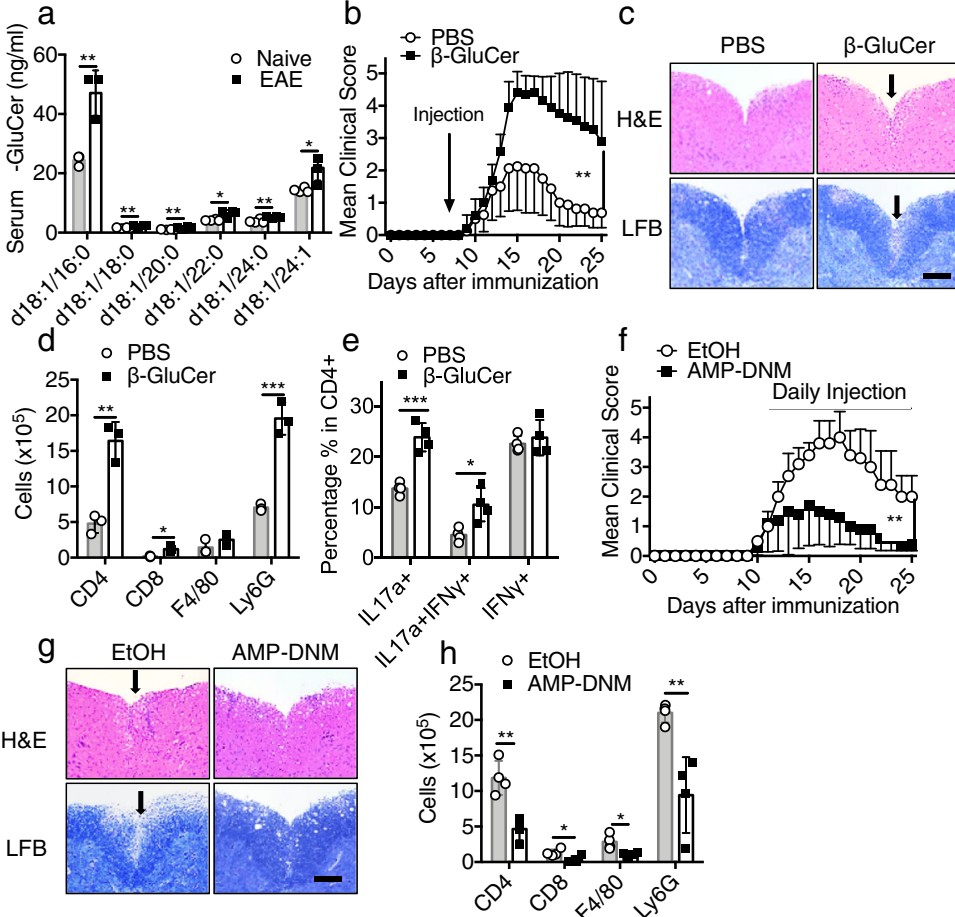

**Fig. 6 β-Glucosylceramide administration promotes EAE. a** Quantification of β-glucosylceramide derivatives obtained from the serum of naive and EAE mice at the peak of the disease, $n = 4$ biological replicates. **b** EAE clinical scores of wild-type mice treated with synthetic β-glucosylceramide (150 μg/mice) or vehicle (PBS) on day 10 after EAE induction, $n = 5$ mice. **c** Hematoxylin and eosin (H&E) staining (upper panels) and Luxol fast blue staining (lower panels) of lumbar spinal cords EAE mice harvested at the peak of disease. Scale bar, 100 μm. Arrows in the upper panel indicate inflammatory cells infiltration, and arrows in the lower panel indicate demyelination area. Representative data are shown for $n = 4$. **d** Absolute numbers of CNS-infiltrating cells were measured at the peak of disease by flow cytometry with indicated antibodies, $n = 3$ biological replicates. **e** Flow cytometry analysis of CD4+ lymphocytes from the brain of EAE mice at the peak of the disease, $n = 4$ biological replicates. **f** EAE clinical score of wild-type mice treated with AMP-DNM (25 mg/kg) or vehicle (EtOH) since the start of EAE symptom, $n = 5$ mice. **g** Hematoxylin and eosin (H&E) staining (upper panels) and Luxol fast blue staining (lower panels) of lumbar spinal cords from EAE mice harvested at the peak of disease. Scale bar, 100 μm. Arrows in the upper panel indicate inflammatory cells infiltration, and arrows in the lower panel indicate demyelination area. Representative data are shown for $n = 4$. **h** Absolute numbers of CNS-infiltrating cells were measured at the peak of disease by flow cytometry with the indicated antibodies, $n = 4$ biological replicates. $*P < 0.05$, $**P < 0.01$ (Two-sided student's $t$ test for **a, d, e, h**). $**P < 0.01$ (two-way ANOVA, **b, d**). Data are represented as mean ± SD. Exact $P$ values for asterisks (from left to right): **a** 0.0016 0.0061 0.0178 0.0103 0.0129 **b** < 0.0001 **d** 0.0024 0.0144 0.0007 **e** 0.00056 0.01513 **f** 0.0033 **h** 0.0023 0.0323 0.0177 0.0057.

cells in the CNS. Similar to other studies showing increased plasma concentrations of β-glucosylceramide in inflammatory and neurological diseases[25,30], plasma concentrations of β-glucosylceramides were increased in plasma from EAE mice compared to naive mice (Fig. 6a). We then investigated the influence of the administration of exogenous β-glucosylceramide on inflammation in the CNS. IV injection of β-glucosylceramide at the onset of disease worsened EAE (Fig. 6b), associated with increased immune cell infiltration and demyelination (Fig. 6c). We also detected more leukocyte infiltration in the CNS of β-glucosylceramide-treated mice (Fig. 6d). Notably, IL-17A+ $T_H17$ cells were substantially increased in β-glucosylceramide-treated mice compared to control mice (Fig. 6e). These data suggest that β-glucosylceramide administration at the onset of disease might directly promote the function of infiltrated $T_H17$ cells in the CNS. As an important control, treatment of *Mincle^{f/f}CD4-Cre* mice with β-glucosylceramide did not affect the development of CNS inflammation (Supplementary

Fig. 6a). On the other hand, daily administration of glucosylceramide synthase inhibitor-AMP-DNM protected mice from EAE progression and reduced immune cell infiltration to the CNS (Fig. 6f–h); however, treatment of *Mincle^{f/f}CD4-Cre* mice with glucosylceramide synthase inhibitor-AMP-DNM failed to further reduce EAE disease (Supplementary Fig. 6b). Notably, blood glucose and serum cholesterol remained at similar concentrations in untreated and AMP-DNM-treated mice (Supplementary Fig. 6c, d). Taken together, these data suggest that the accumulation of β-glucosylceramide during the development of EAE exacerbates the EAE symptoms via activation of Mincle.

To test whether there is an accumulation of β-glucosylceramide in the CNS of EAE mice, we stimulated $T_H17$ cells with lipids extracted from the spinal cord of EAE mice. Lipids extracted from the spinal cord of EAE mice promoted $T_H17$ cell expansion, whereas lipids extracted from the spinal cord of mice treated with glucosylceramide synthase inhibitor-AMP-DNM failed to promote $T_H17$ cells (Fig. 7a).

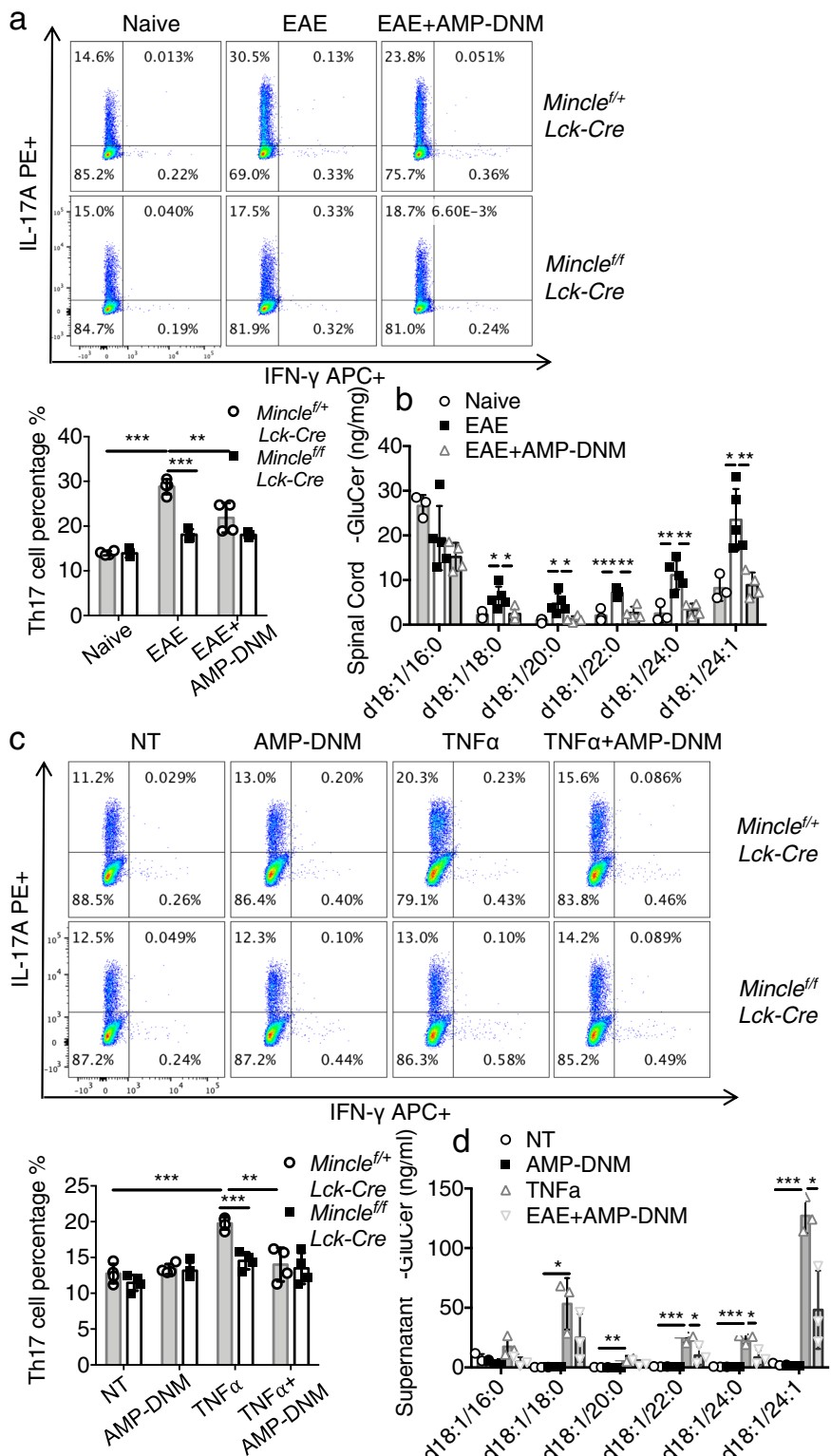

**Fig. 7 Dying oligodendrocytes release β-glucosylceramide to activate TH17 through Mincle. a** Flow cytometric analysis of IL-17A and IFNγ following $T_H17$ polarization with lipids extracted from spinal cord of naive, EAE or AMP-DNM treated EAE mice, $n = 4$ biological replicates. **b** Measurement of β-glucosylceramide from the spinal cord of naive, EAE and AMP-DNM treated EAE mice, $n = 3, 5, 4$ biological replicates. **c** Flow cytometric analysis of IL-17A and IFNγ following $T_H17$ polarization with lipids extracted from supernatants of oligodendrocytes after the indicated treatments, $n = 4$ biological replicates. **d** Measurement of β-glucosylceramide from the oligodendrocyte culture medium under indicated stimulations for 12 h, $n = 3$ biological replicates. $*P < 0.05$, $**P < 0.01$, $***P < 0.001$ (Two sided student's $t$ test). Data are represented as mean ± SD. Exact $P$ values for asterisks (from left to right): **a** < 0.0001 0.00004 0.0091 **b** 0.0376 0.0289 0.00047 0.0065 0.01178 0.0313 0.0193 0.00043 0.0030 0.0056 **c** 0.00013 0.00038 0.00366 **d** 0.0131 0.0055 0.0005 0.0005 0.0001 0.0393 0.0220 0.0190.

Further, β-glucosylceramide concentration was much higher in the spinal cord of EAE mice compared to mice treated with glucosylceramide synthase inhibitor-AMP-DNM (Fig. 7b). Oligodendrocyte death contributes to the pathogenesis of EAE[31,32]. Since TNF is known to drive apoptosis of oligodendrocytes during EAE, we used ex vivo cultures to model TNF-mediated cell death and release of β-glucosylceramide. Lipids extracted from the supernatant of TNF-treated oligodendrocytes promoted $T_H17$ cell expansion (Fig. 7c); this response was blocked by incubation with glucosylceramide synthase inhibitor-AMP-DNM. β-glucosylceramide was indeed accumulated in the supernatant of TNF-treated oligodendrocytes, but decreased in the supernatant of cells treated by the glucosylceramide synthase inhibitor-AMP-DNM (Fig. 7d).

## Discussion

This study reports that sensing of danger signals by Mincle on $T_H17$ cells plays a critical role in promoting CNS inflammation. We demonstrated a T-cell-intrinsic role of *Mincle* in mediating $T_H17$ cell expansion in the CNS for EAE pathogenesis. Mincle was highly expressed in polarized $T_H17$ cells, but not $T_H1$ cells and T-cell-specific deletion of *Mincle* substantially reduced $T_H17$, but not $T_H1$, cell-mediated EAE. Consistently, β-glucosylceramide, an endogenous Mincle ligand, promoted $T_H17$ cell proliferation and blockade of β-glucosylceramide synthesis reduced EAE. Mechanistically, Mincle ligands stimulated the production of IL-1β via ASC-NLRP3-dependent caspase-8 mechanism in activated $T_H17$ cells.

Notably, T-cell-specific *Mincle* deficiency did not affect antigen-dependent priming of $T_H17$ cell or $T_H1$ cell polarization in the spleen, suggesting that IL-1β produced by macrophages and dendritic cells in the peripheral environment is sufficient to support initial $T_H17$ cell priming and expansion. Through $T_H17$ adoptive transfer experiments, while wild-type and *Mincle*-deficient $T_H17$ cells had similar first-wave infiltration into the CNS, *Mincle*-deficient $T_H17$ cells in the CNS failed to proliferate or recruit the second-wave infiltration of inflammatory cells with reduced production of pro-inflammatory cytokines compared with wild-type $T_H17$ cells. In support of this, previous studies utilizing the mouse EAE model have suggested that IL-1β stimulation actually induces the secretion of IL-17A, IFN-γ, GM-CSF, and TNF from $T_H17$ cell–polarized brain-infiltrating cells[33]. Notably, IL-1R is robustly induced during $T_H17$ cell differentiation[34]. Mice deficient in IL-1R have shown significant reductions in EAE disease severity[29,35,36], whereas mice deficient in IL-1Ra, the endogenous soluble IL-1R antagonist, were shown to have a worse disease than wild-type controls. IL-1β stimulation of $T_H17$ cells leads to strong and prolonged activation of the mammalian target of the rapamycin (mTOR) pathway, which has a critical role in cell proliferation and survival and is required for $T_H17$ cell-dependent EAE pathogenesis[37,38]. In support of this, we have previously shown that $Rag1^{-/-}$ mice reconstituted with CD4+ T cells from $IL1$-$β^{-/-}$ mice were protected from the development of EAE[28].

The cord factor (TDM) from HK-Mtb, used as an adjuvant in models of EAE, is a known ligand of Mincle[39]. However, our data suggest that HK-Mtb signaling via Mincle has a limited role in EAE priming or $T_H17$ differentiation. One possible explanation for this discrepancy could be the presence of other T-cell ligands in the HK-Mtb, as HK-Mtb also contains TLR2, TLR4, and TLR9 ligands, such as lipoproteins and lipomannan[40–42]. Intriguingly, both TLR2 and TLR4 signals have been reported to promote $T_H17$ responses and pathogenesis of autoimmune diseases[19,20]. Another possible explanation might be the differences in sensitivity of Mincle compared to other receptors for ligands in HK-Mtb. For example, compared with TLR2, Mincle requires 100–1000× higher concentration of HK-Mtb for activation;

TLR2-elicited responses are stronger than Mincle-dependent responses when stimulated with same concentration of HK-Mtb[43–45]. Further study will be required to clarify which signals are involved in the HK-Mtb triggered $T_H17$ responses.

Our previous work reported that pro-IL-1β in $T_H17$ cells could be processed and secreted in response to stimulation with extracellular ATP in a manner dependent on both ASC and NLRP3[28]. Consistently, the T-cell-intrinsic NLRP3 adaptor ASC was required for the effector stage of EAE. However, that study left one question unresolved: what other danger signals can re-activate $T_H17$ cells in the CNS to stimulate their expansion and conversion towards inflammatory $T_H17$ cells. In this study, we found that β-glucosylceramide, an endogenous Mincle ligand released by dying cells, promoted $T_H17$ cell proliferation in a *Mincle*-dependent manner; blockade of β-glucosylceramide synthesis rescued the mice from EAE. Lipids extracted from the spinal cord of EAE mice promoted $T_H17$ cell expansion, whereas lipid extracts from the spinal cord of mice treated with glucosylceramide synthase inhibitor-AMP-DNM failed to promote $T_H17$ cells. Taken together, this study indicates that sensing of danger signal by Mincle on $T_H17$ cells plays a critical role in promoting CNS inflammation. Importantly, elevated circulating concentrations of β-glucosylceramide were indeed detected in patients with MS[46], implicating the therapeutic potential in targeting the β-glucosylceramide-Mincle axis for patients with MS. Notably, in addition to β-glucosylceramide, SAP130, another endogenous Mincle ligand, has also been implicated in EAE pathogenesis[47]. Moreover, a recent study indicates that microbiota sensing by the Mincle-Syk axis in dendritic cells regulates interleukin-17 and −22 production[48]. While our studies demonstrated the critical role of the T-cell-intrinsic role of *Mincle* in sensing these danger signals released by dying cells, *Mincle* on macrophage or/and dendritic cells might also contribute to EAE pathogenesis.

## Methods

**Mice**. B6 (Cg)-Tg(*Lck-Cd1d1*)1Aben/J, B6 (Cg)-Tg(*CD4*-cre)1Cwi1/BfluJ, B6.129P2-*Lyz2*tm1(cre)Ifo/J and B6J.B6N(Cg)-*Cx3cr1*tm1.1(cre)Jung/J mice (C57BL/6 background) and $Rag1^{-/-}$ were purchased from Jackson Laboratory (stock number 019418, 022071, 004781, 025524 and 002216), $Nlrp3^{-/-}$, $Il1b^{-/-}$, $Caspase1/11^{-/-}$ and *Asc* flox/flox mice were described previously[28]. All of the mice used in this study were female at 10–12 weeks of age, and age-matched littermates were used as experimental groups, at least five mice for each group. These mice were euthanized with carbon dioxide ($CO_2$). Experimental procedures were approved by the Institutional Animal Care and Use Committee of the Cleveland Clinic and mice were housed under specific pathogen-free conditions.

*Generation of* Mincle flox/flox *mice*. The *Mincle flox/flox* mice was generated by Cyagen Biosciences Inc. with the Flp-Frt system. As illustrated in Supplemental Fig. 2a, the "floxed" targeting vector was generated by inserting floxed-neo after exon 5 and a loxP site after exon 2, in order to delete the floxed exon 3–5 by Cre recombinase. The mice carrying Mincle-floxed-neo allele were bred with Flp-Cre mice to delete the neo cassette; the progeny then carried only the loxP sites after exon 2 and exon 5. After backcrossing these mice with C57BL/6 J mice for 8–10 generations, progeny were crossed with mice expressing different cell-specific Cre recombinases to generate cell-specific Mincle-deficient mice. PCR genotyping and ARMS-PCR (amplification refractory mutation system-PCR) were carried out with the following primers: Genotyping (for upstream loxP site, F1: TGGTCAGGATGAGGACACAACAATT, R1: GGGAAGTGGTTAATGCTTTGTGTCC, for downstream loxP site, F2: TGA CTGAACGATA-TCGAATTCCG, R2: GAATTAGGGAAAAGCTGGCAGAA, internal control F2': ACTCCAAFFCCAC-TTATCACC, internal control R2': AT TGTTACCAACTGGGACGACA), ARMS-PCR (F3: CGAATT-CCGAAGTTC CTATTCTCTAG, R3: AGAGTTCCTTGGTCCTATGAGGTTCG).

**Reagents**. Anti-Mincle (1:1000, 1B6) was purchased from MBL. Anti-MCL(1:1000, PA5-102645) was purchased from Thermo Fisher. Anti-ASC (1:1000, N-15-R) was purchased from Santa Cruz Biotechnology. Anti-ASC (1:1000, 2EI-7) and anti-FcRγ(1:1000, 06-727) were purchased from Millipore. Anti-IL-1β (1:1000, AF-401-NA) was purchased from R&D. Anti-Caspase 8 (1:1000, 9429) was purchased from Cell Signaling Technology. Anti-NLRP3 (1:500, H-66) was purchased from SANTA CRUZ Biotechnology. Anti-Caspase 8 (1:1000, 1G12, ALX-804-447-C100) was purchased from Enzo. Anti-NLRP3 (1:500, H-66) was purchased from

SANT CRUZ Biotechnology. Anti-Actin (1:5000, A-2228) was purchased from Sigma. Anti-Ki67 (1:1000, ab15580) was purchased from Abcam. Anti-CD45-APC (1:500, 103112), Anti-F4/80-FITC (1:200, 123108) Anti-mouse-CD25-PE (1:300, PC61, 102008). anti-mouse/human CD44-PE/Cy7 (1:300, IM7, 103030), anti-mouse MCH-II-PerCP/Cy5.5 (1:300, M5/114.15.2, 107626), anti-mouse CD134/OX40-PE/Cy7, OX-86, 119415), anti-mouse GM-CSF-Percp5.5 (1:300, MP1-22E9, 505409), anti-mouse IL-10-PE (1:300, JES5-16E3, 505007) anti-mouse Ki67PE (1:300, 652404), anti-mouse CD3 PE/Cy7(1:300, 100220) and anti-mouse CD8 APC(1:300, 100712) were purchased from Biolegend. Anti-CD4-FITC (1:200, L3T4), Anti-Ly6C-PE (1:300, HK1.4), Anti-IFN-γ-FITC (1:200, XMG1.2), Anti-CD3 (1:1000, 145-2C11), Anti-CD28 (1:1000, 37.51), Anti-IL-4-FITC (1:200, BVD6-24G2) and Anti-FOXP3-PE (1:300, FJK-16S) were purchased from eBioscience. Anti-IL-17A-PE (1:300, 559502), Anti-CD8-PE (1:300, 553041) and Anti-Ly6G-PE (1:300, 1A8) were purchased from BD. Luxol Fast Blue MBS Solution (26681) was purchased from Electron Microscopy Sciences. Caspase-Glo 8 assay system (Kit#G8200) was purchased from Promega. YVAD-fmk (ALX-260-154-R100) and IETD-fmk (550380) were purchased from Enzo and BD. Synthetic β-GluCer [d18:1/C24:1(15Z), C18:0, C16:0, C12:0]) were purchased from Avanti Polar Lipids. TDB was purchased from InvivoGen.

**Induction of EAE and drug treatment**. Active and adoptive transfer EAE model were induced and assessed as previously described[49,50]. Briefly, for the active EAE, mice were immunized with 200 ng MOG35-55 emulsified in CFA (1:1) subcutaneously followed by intraperitoneal injection of pertussis toxin 200 ng on Day0 and Day2. For the adoptive transfer, donor mice were immunized same as active EAE without pertussis toxin and spleen/draining lymph nodes were harvested 10 days after immunization. The cells were in vitro cultured for 5 days with MOG35-55 (15 μg/ml) under either T$_H$1 cell–polarizing conditions (20 ng/ml IL-12, R&D Systems; 2 μg/ml anti-IL-23p19, eBiosciences) or T$_H$17 cell-polarizing conditions (20 ng/ml IL-23, R&D Systems). For drug treatment, mice were treated with β-glucosylceramide (300 μg per mice) by intravenous on day 10 post the immunization. As for AMP-DNM treatment, mice were intraperitoneally injected with AMP-DNM (1 mg/kg) of ethanol dissolved in PBS daily since the onset of the EAE symptom.

**Isolation and differentiation of T cells**. Naive CD4+ T cells were purified with Mojo naive 4 T-cell isolation kit (Biolegend) and percentage of CD4+CD44lo were higher than 98% tested by flow cytometry. Sorted naive CD4 T cells were activated by plate-bound 1 mg/ml anti-CD3 and 1 mg/ml anti-CD28 under differentiation conditions for T$_H$1 (20 ng/ml IL-12, 10 μg/ml anti-IL-4), T$_H$2 (10 ng/ml IL-4, 10 μg/ml anti-IFNγ), T$_H$17 (50 ng/ml IL-6, 2 ng/ml hTGFβ1, 10 μg/ml anti-IFNγ, 10 μg/ml anti-IL-4) and Treg (10 ng/ml hTGFβ1) for 3 days.

**β-Glucosylceramide treatment for inflammasome activation**. Polarized T$_H$17 cells were washed and re-suspended in starvation media (0.1% of serum in RPMI-1640). Then $2 \times 10^6$ T$_H$17 cells were plated on the β-glucosylceramide coated 48-well plates for 12 h. After stimulation, cells were washed and lysed in RIPA lysis buffer. Proteins in cell culture media were concentrated by methanol and chloroform (2:1) method as described[51]. In all, 20 μg of the sample was run on 12% sodium dodecyl–sulfate-polyacrylamide gel electrophoresis (SDS–PAGE) gel and analyzed by western blot.

**Quantitative real-time PCR**. Total RNA was extracted from the spinal cord with TRIzol (Invitrogen) according to the manufacturer's instructions. All gene expression results were expressed as arbitrary units relative to the expression of Actb. Fold induction of gene expression in the spinal cord after EAE induction was determined by dividing the relative abundance of experimental samples by the mean relative abundance of control samples from naive mice. Primers used for real-time PCR are listed in Table 1 in Supplementary files.

**CFSE proliferation assay**. Naive CD4+ T cells were labeled with 5 μM CFSE (Invitrogen) at 37°C for 15 min. Excess dye was washed away by PBS twice. Then cells were cultured in T$_H$17 differentiation media. After 4 days of differentiation, cells were collected and CFSE dilution was assessed by flow cytometry.

**Enzyme-linked immunosorbent assay (ELISA)**. Supernatants were collected for enzyme-linked immunosorbent assay of cytokines with a kit from BioLegend (for IL-17A) or kits from R&D Systems (for all other cytokines). Il-1β levels were assayed by Il-1β (MLB00C) ELISA kit (R&D systems) according to the manufacturer's instructions.

**Western analysis and co-immunoprecipitation assay**. Cells were lysed by lysis buffer (0.5% Triton X-100, 20 mM Hepes pH 7.4,150 mM NaCl, 12.5 mM β-glycerophosphate, 1.5 mM MgCl$_2$, 10 mM NaF, 2 mM dithiothreitol, 1 mM sodium orthovanadate, 2 mM EGTA, 20 mM aprotinin, 1 mM phenylmethylsulfonyl fluoride). 20 μg of protein lysate per lane was run on a 12% SDS-PAGE gel, followed by immune-blotting with different antibodies. Co-immunoprecipitation experiments were performed as described previously[17]. In

brief, cell extracts were incubated overnight with antibodies and protein A beads at 4 °C After incubation, beads were washed four times with lysis buffer, resolved by SDS-PAGE, and analyzed by immunoblotting according to standard procedures.

**Lipid extraction and quantification**. Primary oligodendrocytes were cultured with TNF to induce cell death or pretreated with AMP-DNM to prevent glucosylceramide release. Lipids in the culture medium were extracted with a lipid extraction kit (Biovision) and dried. Lipids from plasma and spinal cord homogenate of EAE mice were also extracted with a lipid extraction kit. The extracted lipids were analyzed with Q-Orbitrap-MS.

**Reporting summary**. Further information on research design is available in the Nature Research Reporting Summary linked to this article.

## Data availability
All data generated in this study are provided in the Supplementary Information and the Source Data file, or are available from the corresponding authors upon request.

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

## Acknowledgements

This study was supported in part by grants from the National Institutes of Health: R01AA023722, to X.L. and L.E.N., P01HL103453, P01AI141350 to X.L, P50AA024333 to L.E.N, P01HL144497 to C.L. R01NS104164 to Z.K.).

## Author contributions

X.L. and Q.R.Z. conceived and coordinated the study, and wrote the manuscript. Q.R.Z. conducted the majority of experiments and analyzed the data. C.J.Z., H.W., W.L., H.Z., K.B., J.J.Z., and X.C. helped with experiments. R.L.Z. performed the mass spectrometry and analyzed the data. Z.K., C.L., R.B., G.D., D.A., T.X., and L.E.N. provided critical discussion, and X.L. and L.E.N. obtained funding.

## Competing interests
The authors declare no competing interests.
