## [Peer Review File · Nature Communications]

TH17 cells promote CNS inflammation by sensing danger signals via MincleREVIEWER COMMENTS

Reviewer #1 (Remarks to the Author):

This study described the pathogenic role of Mincle in CD4⁺ T cells during EAE. The authors demonstrated that Mincle on CD4⁺ T cells promotes Th17 cell polarization and IL-1 β expression by the Th17 cells through Mincle-ASC-NLRP3-Casp8. Furthermore, Mincle responses appeared to be stimulated by b-GluCer as a DAMP during EAE. The source of b-GluCer was suggested to be at least dying oligodendrocytes. Notably, an inhibitor of glucosylceramide synthase ameliorated EAE.

The strength of this study is not only the characterization of the role of Mincle in T cells during EAE, but the authors also identified the connection between Mincle and ASC-NLRP3-Casp8 and suggested a Mincle's endogenous ligand in action. But indeed, intriguing are the part of b-GluCer, as an endogenous ligand of Mincle, and dying oligodendrocytes as a source of b-GluCer. The finding, targeting b-GluCer and Mincle to modify the disease, is fascinating and might have translational potential. The story of the sequence of discoveries is logically described.

Major weaknesses include poorly presented figures, typos and premature edits in text, and the lack of essential information/validation throughout the manuscript. There are numerous issues related to this, but everything should be easy to fix (but essential to be addressed). To list some - Nomenclature of gene/protein names is not reflected. Some letters that need to be superscripted and Italicized are not correctly written. One figure occupies two pages. Figures include random fonts with awkwardly too large or small letters, ending up poor panel organization. One suggestion here is to reorganize figures: Some panels do not need to stay in the main figures too. Scientifically, descriptions of statistics are largely missing on most occasions. For example, qPCR data does not describe what datapoints represent on top of the missing statistical analysis info. Basically, the quality of figures and text must be substantially improved. Last but not least, little information/data on the new Mincle mouse line validation is a significant concern.

Answers to the following questions will be appreciated when accompanied by some data.

- Does the Mincle-Casp8 axis promote cell death? If so, what kind of cell death?
- Which molecule is responsible for processing pro-IL-1 β cleavage in T cells in this setting?

The following items are more about how the results are presented, but authors' reply is asked for scientific/data validation.

- Bands in WB data should not be cropped without any sizes, as shown in the current version. This website is helpful (<http://jbcresources.asbmb.org/collecting-and-presenting-data>).
- Fig. 2A. Please do re-make this panel from scratch. Readers will have absolutely no idea which exons they are and how the WT gene looks, etc. This panel is not helpful.
- Relating to Fig. 2A. More detailed information on the new Mincle mouse line generation is necessary. Exactly which part of the gene was targeted, how you do genotyping (and primer sequences).
- More info about the validation of the Mincle(f/f) line is essential. Southern hybridization looked OK? Is Mincle deletion truly works in a Cre-specific way?
- Fig. 3H key. Recipients are described as "WT" but aren't they Rag1 KO?
- Fig. 3J does not make much sense and is helpful to be there. It even does not have a key in the panel.
- Line 156 says that Fig. 3D is for Th17 cells, and so on. But it seems the description was mistakenly switched between Fig. 3D and 3E.

Reviewer #2 (Remarks to the Author):

The study of Quanri Zhang and coworkers shows an unexpected intrinsic role for the C-type lectin receptor Mincle (Clec4e) in Th17 cells. The work nicely and straightforward suggests that sensing of danger signal (β -glucosylceramide) by Mincle on TH17 cells plays a critical role in experimental autoimmune encephalomyelitis (EAE). The experimental methodology used is rather robust and the findings interesting. Notwithstanding, I have several comments:

-EAE is induced by inoculation of MOG35-55 emulsified in CFA, which contains trehalose 6,6'-dimycolate (TDM). TDM is one of the best characterized ligands for Mincle. The authors would like to discuss whether the use of this adjuvant impacts on early primed Th17 cells in vivo, since they demonstrate in supplementary figure 4 (Fig S4) that TDB indeed has an effect during Th17 priming.

-Similarly, it would be of interest to determine the kinetics of Mincle expression upon TCR stimulation in vitro (is Mincle expressed 24 or 48h after stimulation?(Figure 4). Moreover, it would be also important to determine the kinetics of Mincle expression during in vivo priming. Although the authors conclude that Mincle does not impact on Th17 priming, figure 7A and S4 suggest the opposite, since lipids extracted from spinal cord or TDB promotes Th17 polarization.

-Figure 1 and Supp 1. According to what has been published so far, Clec4d (dectin 3, MCL) is needed for Clec4e expression, which is inducible by some stimuli (J Immunol June 1, 2015, 194 (11) 5366-5374). Clec4e and Clec4d form a heterodimer that associates with FcRg to form a functional complex (Eur. J. Immunol. 2013. 43: 3167–3174). GSE data shown by the authors indicate that Th17 cells do not express Clec4d. Regarding the data presented by the authors, they should consider that the anti-Clec4e 1B6 antibody can also recognize Clec4d, although with lower affinity (Immunity 38, 1050–1062, 2013). Clec4e is dispensable for Clec4d expression, at least in myeloid cells. Therefore, some residual signal in mincle-deficient cells revealed with 1B6 could be expected, but it does not seem to be the case (Figure 1B). This result does not fit with the detection of some dectin 3 mRNA expression (Figure S1D). The authors would like to clarify these issues by directly determining Clec4d expression using a specific antibody (3A4 antibody; Eur J Immunol. 2016 Feb; 46(2): 381–389). It would be also important to determine FcRg expression in Th17 cells, and whether engagement of Mincle results in syk phosphorylation (as expected). This information will help to understand how, and which is the function of this CLR in Th17 cells.

Figure 3J. Please, add a proper legend to this panel figure indicating which fluorescence represent each color.

Figure 5. These WB analysis might require band quantification. For example, it is not completely clear if GlcCer treatment increases NLRP3 expression in WT cells. It will be of interest if the authors show simultaneously, and in the same blot the two bands corresponding to Pro-IL1b and IL1b (for example:

Nat Immunol. 2014 Aug; 15(8): 738-48). This will clarify whether GlcCer induces pro-IL1b expression, pro-IL1b processing or, most probably, both.

Minor:

-The authors should use either GlcCer or Glucer as an abbreviation.

-Line 143 Gm-csf

-Please, review carefully the manuscript, there are some typos/ grammar errors. For example: Figure 4. Mincle activation promote's' (or promoted) TH17 differentiation; Supplementary Fig 4. TDB promote's' (or promoted) Th17 differentiation through Mincle signal

We thank the reviewers for their careful and positive review of our manuscript. We have been able to respond to each of the criticisms and questions by including new data and clarifying our presentation of the data. Below please find a point by point response to reviews. Major changes in the manuscript are highlighted in yellow; however, we did not track all the editorial changes related to nomenclature, italics, etc.

Reviewer #1 (Remarks to the Author):

This study described the pathogenic role of Mincle in CD4+ T cells during EAE. The authors demonstrated that Mincle on CD4+ T cells promotes Th17 cell polarization and IL-1b expression by the Th17 cells through Mincle-ASC-NLRP3-Casp8. Furthermore, Mincle responses appeared to be stimulated by b-GluCer as a DAMP during EAE. The source of b-GluCer was suggested to be at least dying oligodendrocytes. Notably, an inhibitor of glucosylceramide synthase ameliorated EAE.

The strength of this study is not only the characterization of the role of Mincle in T cells during EAE, but the authors also identified the connection between Mincle and ASC-NLRP3-Casp8 and suggested a Mincle's endogenous ligand in action. But indeed, intriguing are the part of b-GluCer, as an endogenous ligand of Mincle, and dying oligodendrocytes as a source of b-GluCer. The finding, targeting b-GluCer and Mincle to modify the disease, is fascinating and might have translational potential. The story of the sequence of discoveries is logically described.

Major weaknesses include poorly presented figures, typos and premature edits in text, and the lack of essential information/validation throughout the manuscript. There are numerous issues related to this, but everything should be easy to fix (but essential to be addressed). To list some - Nomenclature of gene/protein names is not reflected. Some letters that need to be superscripted and Italicized are not correctly written. One figure occupies two pages. Figures include random fonts with awkwardly too large or small letters, ending up poor panel organization. One suggestion here is to reorganize figures: Some panels do not need to stay in the main figures too. Scientifically, descriptions of statistics are largely missing on most occasions. For example, qPCR data does not describe what datapoints represent on top of the missing statistical analysis info. Basically, the quality of figures and text must be substantially improved. Last but not least, little information/data on the new Mincle mouse line validation is a significant concern.

RESPONSE: We apologize for the poor presentation of the data in manuscript and figures. We have reorganized the figures and improved the quality of figures and text as suggested.

Answers to the following questions will be appreciated when accompanied by some data.

- Does the Mincle-Casp8 axis promote cell death? If so, what kind of cell death?

RESPONSE: In order to answer this excellent question, we conducted new experiments using the Annexin V/Propidium Iodide flow cytometry assay in Th17 cells to assess cell death upon Mincle-Casp8 activation by β -GluCer. The percentage of dead cells (Annexin V⁺, early apoptotic or Annexin V⁺/PI⁺, late apoptotic, Necrotic) was not increased in response to challenge with β -GluCer when compared to non-treated cells. Cells treated with PMA and ionomycin as a positive control exhibited increases in both early apoptotic and late apoptotic/necrotic populations. This data is now included as Supplementary Figure 6C.

- Which molecule is responsible for processing pro-IL-1b cleavage in T cells in this setting?

RESPONSE: Based on previous literature and our own data, we have concluded that caspase-8 contributes to the processing of pro-IL-1 β . It has been reported that in the myeloid cells, Caspase-8 functions as a non-canonical inflammatory caspase to process pro-IL1 β ¹⁻³. Caspase-8 has also been found exert an important non-apoptotic role in T cells¹. In our previous work, we reported that Th17 intrinsic ASC-NLRP3-Caspase-8 signal is required for pro-IL1 β processing⁴. In the current study, caspase-8 was associated with ASC and NLRP3 in pull-down assays in response to Mincle activation (Fig. 5H). Further evidence presented to the reviewers is shown in the western blot above demonstrating that caspase-8 inhibitor decreased β -GluCer stimulated processing of IL1 β . Based on this accumulated evidence, we concluded that Mincle signals to promote pro-IL1 β processing via the ASC-NLRP3-Caspase-8 axis.

The following items are more about how the results are presented, but authors' reply is asked for scientific/data validation.

- Bands in WB data should not be cropped without any sizes, as shown in the current version. This website is helpful (<http://jbcresources.asbmb.org/collecting-and-presenting-data>).

RESPONSE: Thank you for your valuable comment. Molecular weights are now indicated for the western blots and larger, un-cropped portions are presented in Supplementary Figures.

- Fig. 2A. Please do re-make this panel from scratch. Readers will have absolutely no idea which exons they are and how the WT gene looks, etc. This panel is not helpful.

- Relating to Fig. 2A. More detailed information on the new Mincle mouse line generation is necessary.

Exactly which part of the gene was targeted, how you do genotyping (and primer sequences).

RESPONSE: Thank you for the suggestions. We have re-drawn the panel for the Mincle flox strategy (Fig 2A and Supplementary Fig. 2) and added the sequences of genotyping primers and genotyping results of Mincle^{+/+}, Mincle^{f/+}, Mincle^{f/f} in the Methods section.

- More info about the validation of the *Mincle(f/f)* line is essential. Southern hybridization looked OK? Is *Mincle* deletion truly works in a Cre-specific way?

RESPONSE: In order to address the reviewers question, we checked *Mincle* expression in T_H17 cells and bone marrow macrophages (BMM) from *Mincle^{f/f}* and *Mincle^{f/f} CD4-cre* mice. *Mincle* expression was not detectable in T_H17 cells from the *Mincle^{f/f} CD4-cre* but was readily detected in BMM. This data is now included in Fig. 2A.

- Fig. 3H key. Recipients are described as "WT" but aren't they Rag1 KO?

We have corrected the labeling to *Rag1^{-/-}* in Fig. 3H.

- Fig. 3J does not make much sense and is helpful to be there. It even does not have a key in the panel.

We are thankful for the reviewer's comments. In order to make the figure clearer, we moved this panel and its key to Supplementary Figure 3F.

- Line 156 says that Fig. 3D is for Th17 cells, and so on. But it seems the description was mistakenly switched between Fig. 3D and 3E.

We apologize for the mistakes and we have corrected the panels for Fig. 3D and 3E.

Reviewer #2 (Remarks to the Author):

The study of Quanri Zhang and coworkers shows an unexpected intrinsic role for the C-type lectin receptor *Mincle* (*Clec4e*) in Th17 cells. The work nicely and straightforward suggests that sensing of danger signal (β -glucosylceramide) by *Mincle* on TH17 cells plays a critical role in experimental autoimmune encephalomyelitis (EAE). The experimental methodology used is rather robust and the findings interesting. Notwithstanding, I have several comments:

-EAE is induced by inoculation of MOG35-55 emulsified in CFA, which contains trehalose 6,6' dimycolate (TDM). TDM is one of the best characterized ligands for *Mincle*. The authors would like to discuss whether the use of this adjuvant impacts on early primed Th17 cells in vivo, since they demonstrate in supplementary

figure 4 (Fig S4) that TDB indeed has an effect during Th17 priming.

RESPONSE: We thank the reviewer for this valuable comment. We have shown in the paper (Supplementary Figure 3 B-E) that the heat-killed Mycobacterium-Tuberculosis (HK-Mtb) did not change the percentage, viability, activation and subpopulation of splenic CD4 T cells during EAE priming stage. Additionally, we tested the influence of HK-Mtb on the polarization of Th17 cells. In new data, shown in the Figure above, addition of HK-Mtb promoted the percentage of IL-17A+ cells in both *Mincle*^{+/-} and *Mincle*^{-/-} cells, while challenge with β-GluCer increased IL-17A+ cells on in *Mincle*^{+/-} cells. Taken together, these data suggest that HK-Mtb may affect Th17 via other, Mincle independent, mechanisms.

ACTION: We have now added the following comments to the Discussion to clarify this issue:

The cord factor (TDM) from heat killed-Mycobacterium Tuberculosis(HK-Mtb) is a known ligand of Mincle⁵. However, our data suggest that HK-Mtb has a limited role in EAE priming or Th17 differentiation via Mincle signal. One possible explanation for this discrepancy could be the presence of other T cell ligands in the HK-Mtb, as HK-Mtb also contains TLR2, TLR4 and TLR9 ligands, such as lipoproteins and lipomannan⁶⁻⁸. Intriguingly, both TLR2 and TLR4 signals have been reported to promote Th17 responses and pathogenesis of autoimmune diseases^{9,10}. Another possible explanation might be the differences in sensitivity of Mincle compared to other receptors for ligands in HK-Mtb. For example, compared with TLR2, Mincle requires 100-1000x higher concentration of of HK-Mtb for activation; TLR2 elicited responses are stronger than Mincle when stimulated with same concentration of HK-Mtb¹¹⁻¹³. Further study will be required to clarify which signal is involved in the HK-Mtb triggered Th17 response.

-Similarly, it would be of interest to determine the kinetics of Mincle expression upon TCR stimulation in vitro (is Mincle expressed 24 or 48h after stimulation?(Figure 4). Moreover, it would be also important to determine the kinetics of Mincle expression during in vivo priming. Although the authors conclude that Mincle does not impact on Th17 priming, figure 7A and S4 suggest the opposite, since lipids extracted from spinal cord or TDB promotes Th17 polarization.

RESPONSE: We thank the reviewer for this question. We detected Mincle protein expression as early as 48 h of the *in vitro* polarization. Further, Mincle protein could be detected in splenic CD4 T cells isolated from MOG immunized mice by day 6. This data has been added to Supplement Figure 1E.

-Figure 1 and Supp 1. According to what has been published so far, Clec4d (dectin 3, MCL) is needed for Clec4e expression, which is inducible by some stimuli (J Immunol June 1, 2015, 194 (11) 5366-5374). Clec4e and Clec4d form a heterodimer that associates with FcRg to form a functional complex (Eur. J. Immunol. 2013. 43: 3167–3174). GSE data shown by the authors indicate that Th17 cells do not express Clec4d. Regarding the data presented by the authors, they should consider that the anti-Clec4e 1B6 antibody can also recognize Clec4d, although with lower affinity (Immunity 38, 1050–1062, 2013). Clec4e is dispensable for Clec4d expression, at least in myeloid cells. Therefore, some residual signal in mincle-deficient cells revealed with 1B6 could be expected, but it does not seem to be the case (Figure 1B). This result does not fit with the detection of some dectin 3 mRNA expression (Figure S1D). The authors would like to clarify these issues by directly determining Clec4d expression using a specific antibody (3A4 antibody; Eur J Immunol. 2016 Feb; 46(2): 381–389)

RESPONSE: Thank you for your valuable comments. To address your concern, as you suggested, we directly probed for MCL in western blots using the Invitrogen PA5-102645 antibody. MCL was not detected in Th17 cells by western blot; bone marrow macrophages (BMM) were used as a positive control. Therefore, interference of MCL protein expression on Mincle western blot should be limited. This data has been added to Supplemental Figure 1F. We also rechecked the expression of MCL (dectin-3) mRNA (Supplementary Fig. 1D); induction of MCL mRNA was much lower than induction of Mincle mRNA in Th17 cells, consistent with the western blot data.

-It would be also important to determine FcRg expression in Th17 cells, and whether engagement of Mincle results in syk phosphorylation (as expected). This information will help to understand how, and which is the function of this CLR in Th17 cells.

RESPONSE: To answer this question, Th17 cells were stimulated with β -GluCer. We then pulled down Mincle and observed that FcR γ and Syk were in the Mincle immunoprecipitate. These data indicate that both FcR γ and Syk are engaged in Th17 Mincle signaling. These data are now included in Supplementary Figure 5B.

Figure 3J. Please, add a proper legend to this panel figure indicating which fluorescence represent each color.

We apologize for the missing information. We have added the details for this panel to the legend.

5. These WB analysis might require band quantification. For example, it is not completely clear if GlcCer treatment increases NLRP3 expression in WT cells. It will be of interest if the authors show simultaneously, and in the same blot the two bands corresponding to Pro-IL1 β and IL1 β (for example:

Nat Immunol. 2014 Aug;15(8):738-48). This will clarify whether GlcCer induces pro-IL1 β expression, pro-IL1 β processing or, most probably, both.

RESPONSE: We have added the quantification to western blot band. We cannot show pro- and cleaved IL1 β in the same blot as we measured cleaved IL1- β in the cell culture media and pro-IL1 β in the cell lysates. However, we have now included western blot images from the pro-IL1 β containing lysates in order to address the reviewer's concern.

Minor:

-The authors should use either GlcCer or Glucer as an abbreviation.

We apologize for the confusion, we have made the labeling consistent.

-Line 143 Gm-csf

-Please, review carefully the manuscript, there are some typos/ grammar errors. For example: Figure 4.

Mincle activation promote's' (or promoted) TH17 differentiation; Supplementary Fig 4. TDB promote's' (or promoted) Th17 differentiation through Mincle signal

We apologize for the mistakes, and we have carefully reviewed the manuscript to avoid the errors.

1. Gringhuis SI, Kaptein TM, Wevers BA, et al. Dectin-1 is an extracellular pathogen sensor for the induction and processing of IL-1 β via a noncanonical caspase-8 inflammasome. *Nat Immunol.* 2012;13(3):246-254.
2. Bossaller L, Chiang PI, Schmidt-Lauber C, et al. Cutting edge: FAS (CD95) mediates noncanonical IL-1 β and IL-18 maturation via caspase-8 in an RIP3-independent manner. *J Immunol.* 2012;189(12):5508-5512.
3. Antonopoulos C, El Sanadi C, Kaiser WJ, et al. Proapoptotic chemotherapeutic drugs induce noncanonical processing and release of IL-1 β via caspase-8 in dendritic cells. *J Immunol.* 2013;191(9):4789-4803.
4. Martin BN, Wang C, Zhang CJ, et al. T cell-intrinsic ASC critically promotes T(H)17-mediated experimental autoimmune encephalomyelitis. *Nat Immunol.* 2016;17(5):583-592.
5. Ishikawa E, Ishikawa T, Morita YS, et al. Direct recognition of the mycobacterial glycolipid, trehalose dimycolate, by C-type lectin Mincle. *J Exp Med.* 2009;206(13):2879-2888.
6. Drage MG, Pecora ND, Hise AG, et al. TLR2 and its co-receptors determine responses of macrophages and dendritic cells to lipoproteins of *Mycobacterium tuberculosis*. *Cell Immunol.* 2009;258(1):29-37.
7. Pecora ND, Gehring AJ, Canaday DH, et al. *Mycobacterium tuberculosis* LprA is a lipoprotein agonist of TLR2 that regulates innate immunity and APC function. *J Immunol.* 2006;177(1):422-429.

8. Bafica A, Scanga CA, Feng CG, et al. TLR9 regulates Th1 responses and cooperates with TLR2 in mediating optimal resistance to *Mycobacterium tuberculosis*. *J Exp Med*. 2005;202(12):1715-1724.
9. Reynolds JM, Martinez GJ, Chung Y, et al. Toll-like receptor 4 signaling in T cells promotes autoimmune inflammation. *Proc Natl Acad Sci U S A*. 2012;109(32):13064-13069.
10. Reynolds JM, Pappu BP, Peng J, et al. Toll-like receptor 2 signaling in CD4(+) T lymphocytes promotes T helper 17 responses and regulates the pathogenesis of autoimmune disease. *Immunity*. 2010;32(5):692-702.
11. Underhill DM, Ozinsky A, Smith KD, et al. Toll-like receptor-2 mediates mycobacteria-induced proinflammatory signaling in macrophages. *Proc Natl Acad Sci U S A*. 1999;96(25):14459-14463.
12. Schoenen H, Bodendorfer B, Hitchens K, et al. Cutting edge: Mincle is essential for recognition and adjuvanticity of the mycobacterial cord factor and its synthetic analog trehalose-dibehenate. *J Immunol*. 2010;184(6):2756-2760.
13. Kerscher B, Willment JA, Brown GD. The Dectin-2 family of C-type lectin-like receptors: an update. *Int Immunol*. 2013;25(5):271-277.

REVIEWER COMMENTS

Reviewer #1 (Remarks to the Author):

The authors thoroughly answered questions by reviewers, and the manuscript is greatly improved. However, I still thought that information for the new Mincle-flox mouse line is largely missing. I would like to ask for the following issues.

Please indicate where the genotyping primers anneal for the Mincle-flox mouse genotyping in Methods with words. Supp Fig. 2A schematics indicates it by a box with a broken line, but it is not helpful. Please also update the Supp Fig. 2A with a more enlarged diagram.

PCR data in Supp Fig. 2B is not helpful because there is no description about which part of the gene was amplified. Please indicate primer positions in a diagram, as well as adding the description where they anneal in the Methods.

Also, please describe how the Mincle-flox mice were generated in a subsection of Methods. The general mouse generation info is completely missing.

Reviewer #2 (Remarks to the Author):

My concerns have been addressed in the revised manuscript. I would like to congratulate the authors for their work.

Point by point response

Reviewer #1 (Remarks to the Author):

The authors thoroughly answered questions by reviewers, and the manuscript is greatly improved. However, I still thought that information for the new *Mincle*-flox mouse line is largely missing. I would like to ask for the following issues.

Please indicate where the genotyping primers anneal for the *Mincle*-flox mouse genotyping in Methods with words. Supp Fig. 2A schematics indicates it by a box with a broken line, but it is not helpful. Please also update the Supp Fig. 2A with a more enlarged diagram.

PCR data in Supp Fig. 2B is not helpful because there is no description about which part of the gene was amplified. Please indicate primer positions in a diagram, as well as adding the description where they anneal in the Methods.

RESPONSE: We have revised Supplemental Figure 2 as suggested to include more specific details

Also, please describe how the *Mincle*-flox mice were generated in a subsection of Methods. The general mouse generation info is completely missing.

RESPONSE: We have added the following detailed method to the Methods sections.

Generation of *Mincle* flox/flox mice

The *Mincle* flox/flox mice was generated by Cyagen Biosciences Inc with the Loxp-Frt system. As illustrated in **Supplemental Fig.2a**, the “floxed” targeting vector was generated by inserting floxed-neo after exon 5 and a loxP site after exon 2, in order to delete the floxed exon 3-5 by Cre recombinase. The mice carrying *Mincle*-floxed-neo allele were bred with *Ella*-Cre mice to delete the neo cassette; the progeny then carried only the loxP sites after exon 2 and exon 5. After backcrossing of these mice with C57BL/6J mice for 8-10 generations, progeny were crossed with mice expressing different cell-specific Cre recombinases to generate cell specific *Mincle* deficient mice. PCR genotyping and ARMS-PCR (amplification refractory mutation system-PCR) were carried out with the following primers: Genotyping (for upstream loxP site, F1: CTGGTCAGGATGAGGACACAACAATT, R1: GGAAGTGGTTAATGCT-TTGTGTCC, for downstream loxP site, F2: TGAAGTGAACGATATCGAATTCCG, R2: GAATTAGG-GAAAAGCTGGCAGAA,), ARMS-PCR (F3: CGAATTCCGA-AGTTCCTATTCTCTAG, R3: AGAGT-TCCTTGGTCCTATGAGGTTTCG)

REVIEWERS' COMMENTS

Reviewer #1 (Remarks to the Author):

The authors added information on the Mincle flox mouse line, which is very appreciated. Congratulations on the nice work.